

# Using a golf specific functional movement screen to predict golf performance in collegiate golfers

Min Shi[1,2], Hua Wu[1], Hui Ruan[1], Dan Xu[3], Libo Deng[4] and Shibo Pang[5]

[1] Faculty of Physical Education, Hainan Normal University, Haikou, Hainan Province, China
[2] Hainan College of Software Technology, Qionghai, China
[3] Hainan Provincial Sports Academy, Haikou, China
[4] Yulin Normal University, Yulin, China
[5] Hainan College of Economics and Business, Haikou, China

## ABSTRACT

**Background:** This study aims to examine the relationship between functional movements and golf performance using the Golf Specific Functional Movement Screen (GSFMS).

**Methods:** This cross-sectional study included a total of 56 collegiate golfers (aged $20.89 \pm 0.99$ years, height of $174.55 \pm 7.76$ cm, and weight $68.48 \pm 9.30$ kg) who met the criteria, and were recruited from Hainan Normal University in June 2022. The participants' golf motor skills (1-yard putt, 10-yard putt, 25-yard chip, 130/100-yard set shot, driver, and 9-hole stroke play) were tested and the GSFMS (*e.g.*, pelvic tilt, pelvic rotation, and torso rotation) was used.

**Results:** There were significant weak or moderate correlations between the variables. Furthermore, a multiple linear regression analysis found that pelvic rotation and lower-body rotation abilities can significantly predict golf skill levels, which collectively explain 31.2% of the variance in golf skill levels among collegiate golfers (Adjusted $R^2 = 0.312$, F = 2.663, $p < 0.05$). Standardised $\beta$ values indicate that pelvic rotation ($\beta = 0.398$) has a more substantial impact on golf skill levels than lower-body rotation ($\beta = 0.315$).

**Conclusions:** This study found the weak to moderate correlations between the GSFMS and golf performance, and pelvic rotation and lower-body rotation abilities, thus predicting golf skills. Our findings provide novel insights into the relationship between functional abilities and comprehensive skill performance within the context of the Gray Cook's Movement Pyramid model, and provide theoretical support and practical reference for collegiate golf motor-skill learning and sports injury prevention.

Corresponding authors
Hua Wu, wuhua0049@gmail.com
Hui Ruan, huiruan@kkumail.com

## INTRODUCTION

Golf is an increasingly popular sport, with a growing number of people participating worldwide, both for recreational enjoyment and in professional competitions. The global number of golfers increased from 61 to 66.6 million from 2016 to 2021 (*The R & A,*

2021). However, golf is a complex sport in which excellent motor skills, technical proficiency, mental acuity, and physical prowess are key to achieving optimal performance on the course (*Sheehan, Bower & Watsford, 2022*). Golf swings are often regarded as one of the most intricate and challenging movements in sports (*Dillman & Lange, 1994*). A successful golf swing requires harmonious synchronization of various body parts and muscle groups, underscoring the significance of an efficient movement pattern. To increase distance, factors such as swing technique, clubhead speed, and physical strength are considered influencing elements of golf performance (*Fletcher & Hartwell, 2004*).

While golf is generally considered to be low in intensity and carries a lower risk of injury compared to ball sports, it is not without its potential for injuries. Each year, approximately 60% of professional and 40% of amateur players experience discomfort or injury (*Brandon & Pearce, 2009*); injuries among professional players are often associated with overuse, whereas injuries among amateur golfers are frequently linked to improper swing mechanics (*Me, 1993*). The commonly used Functional Movement Screen (FMS) is widely employed to identify potential injury risks, such as weak links in movement patterns and physical mobility restrictions (*Cook, 2011*; *Frost et al., 2012*).

The FMS is a body functional movement assessment tool designed by *Cook (2011)*. and includes the deep squat, hurdle step, in-line lunge, shoulder mobility, active straight leg raise, trunk stability push, and rotary stability seven tests. However, the specific characteristics of different sports dictate the requirement for specialised screening to enhance sensitivity in detecting abnormal patterns (*Gould et al., 2017*). The Titleist Performance Institute (TPI) has developed the Golf Specific Functional Movement Screen (GSFMS) (*Rose, 2003*) that identifies potential issues that athletes may encounter during golf swings by conducting 16 specific tests and assessments under particular postures. These include body flexibility, core stability, balance control, and range of motion. The goal is to optimise athletes' swing techniques and overall performance.

Gray Cook's Movement Pyramid model posits three levels: functional movement as the foundation, assessing basic movement patterns; functional performance in the middle, evaluating qualities like speed and explosiveness; and functional skills at the top, involving advanced techniques and coordination, typically assessed through specific skill tests. (*Cook, 2010*). According to Cook, issues at the foundational level can impact higher layers of movement capabilities. Previous research has found associations between functional movement and aspects such as physical performance, movement performance (*Fitton Davies et al., 2022*; *Zhang et al., 2022*), and the occurrence of sports injuries (*Chorba et al., 2010*; *Dïnç & Arslan, 2020*). However, the relationship between functional movement and movement performance lacks consensus (*Stapleton et al., 2021*; *Bennett et al., 2022*).

Only four prior studies exist on golf. *Gulgin, Schulte & Crawley (2014)* explored the connection between functional movement limitations and golf swing faults, but did not utilise the complete GSFMS tests, although all these functional movements are relevant to swing performance. Additionally, golf swing faults do not directly reflect golf performance. *Warren, Smith & Chimera (2015)* investigated the correlation between FMS and sports injuries among amateur golfers. *Speariett & Armstrong (2020)* investigated the connection between GSFMS composite scores and golf performance, utilising metrics such as

handicap, clubhead speed, side accuracy, ball speed, peak pelvis rotation speed, swing sequence, and common swing faults. Their evaluation encompasses aspects such as body movement, coordination, power transfer, and technical nuances, which may be more abstract than specific golf skills (*Speariett & Armstrong, 2020*). *Gould et al. (2021)* have utilised their self-developed Golf Movement Screen to explore the correlation between functional movement and biomechanical indicators of swing action. Therefore, it can be concluded from existing studies that the relationship between functional movement and golf performance is multidimensional and complex. Different functional movement screening tools and assessment indicators offer various pathways and perspectives for exploring this relationship. This study aimed to deepen our understanding of the relationship between functional movement and overall golf performance. GSMFS was used to assess functional movements, and the national standard of student sports skill rating by age and sport was used to assess golf performance (*Ministry of Education of the People's Republic of China, 2019*). The golf performance indicators used in this study, compared with those used in previous research, places greater emphasis on integration of golf into with sports competitions. It encompasses various skills involved in playing a game, including short putts, long putts, chipping, iron shots, wood shots, and various skills in on-course practice. We hypothesised that golf special functional movements would have a moderate correlation with golf performance. By focusing on specific shot techniques and competitive performance in the study, we conducted a comprehensive analysis of the impact of different functional movements on golf skills. This will enhance our understanding of the connection between functional movements and golf. performance.

## MATERIALS AND METHODS

### Experimental approach to the problem

A cross-sectional study was designed to determine the correlation between functional movement patterns assessed by the GSFMS and overall golf performance in collegiate golfers. This study adhered to the STROBE guidelines (*Vandenbroucke et al., 2007*) and was conducted at Hainan Normal University (Haikou Province, China) in June 2022.

### Participants

Based on the research findings of *Speariett & Armstrong (2020)* and using the formula by *Sharma et al. (2020)* Sample size (N) = $\frac{(z_{1-\alpha/2})^2 * (\sigma)^2}{(d)^2}$, with a 95% confidence interval (CI) and an allowable error of 1, the calculated sample size required was 56 individuals. The inclusion criteria were as follows: (1) good physical health and (2) golf major students who had not engaged in golf before university and volunteered to participate in the test. The exclusion criteria involved individuals who had not participated in golf within the past 6 months due to injury.

The study included 56 collegiate golfers from Hainan Normal University who specialised in golf sports and management; they systematically studied and practiced golf theory and skills for 2–3 years, and they usually practiced golf 2–3 times a week on average,

**Table 1 Basic information of participants.**

|  | N | M ± SD |
|---|---|---|
| Age (Years) | 56 | 20.89 ± 0.99 |
| Height (cm) | 56 | 174.55 ± 7.76 |
| Weight (kg) | 56 | 68.48 ± 9.30 |
| BMI | 56 | 22.41 ± 2.06 |
| 1-yard putt | 56 | 9.25 ± 1.46 |
| 10-yard putt | 56 | 7.71 ± 2.27 |
| 25-yard chip shot | 56 | 5.57 ± 2.62 |
| 130/100-yard pitch | 56 | 7.98 ± 1.78 |
| Driving | 56 | 6.15 ± 2.04 |
| One round of 9-hole stroke play score | 56 | 23.55 ± 3.95 |
| Total skills score | 56 | 60.21 ± 8.76 |
| PT | 56 | 1.68 ± 0.54 |
| PR | 56 | 1.45 ± 0.83 |
| TR | 56 | 0.77 ± 0.43 |
| ODS | 56 | 2.25 ± 1.00 |
| TT | 56 | 0.82 ± 0.39 |
| 90/90 | 56 | 2.32 ± 1.52 |
| SLB | 56 | 0.18 ± 0.43 |
| LT | 56 | 1.86 ± 0.52 |
| LQR | 56 | 3.71 ± 0.71 |
| STR | 56 | 1.57 ± 0.74 |
| BWLE | 56 | 0.41 ± 0.65 |
| CR | 56 | 1.64 ± 0.70 |
| FR | 56 | 1.68 ± 0.69 |
| WH | 56 | 2.00 ± 0.00 |
| WF | 56 | 1.20 ± 0.98 |
| WE | 56 | 1.18 ± 0.99 |
| GSFMS | 56 | 24.71 ± 4.52 |

Note:
Abbreviations: PT, Pelvic Tilt Test; PR, Pelvic Rotation Test; TR, Torso Rotation Test; ODS, Overhead Deep Squat Test; TT, Toe Touch Test; 90/90, 90/90 Test; SLB, Single Leg Balance Test; LT, Lat Test; LQR, Lower Quarter Rotation Test; STR, Seated Trunk Rotation Test; BWLE, Bridge with Leg Extension Test; CR, Cervical Rotation Test; FR, Forearm Rotation (Pronation/Supination) Test; WH, Wrist Hinge Test; WF, Wrist Flexion Test; WE, The Wrist Extension Test.

about 3 h each time. The basic information of the participants is presented in Table 1. This study adhered to the Declaration of Helsinki and was approved by the Ethics Committee of the Hainan Provincial Sports Academy (No. GT-QM-02). All participants were informed of the details and procedures of the test and provided written informed consent.

## Procedures

The experiment was conducted in the Golf Swing Technique Training Lab of the Faculty of Physical Education at Hainan Normal University. The study encompassed both the GSFMS and golf performance testing, with all assessments completed within 1 week. The Greenjoy Q9 simulator (Shenzhen Greenjoy Technology, Shenzhen, China) was used

as the golf skill testing venue, coupled with the 2020 Cloud Eye binocular high-speed camera sensor certified by the European Union Conformité Européenne and Institution of Civil Engineers, and it features an automatic putting system and an extra high definition high-frame-rate camera. A separate area served as the GSFMS testing site. Data collection was performed by an assistant who had received GSFMS training. Without warming up, the participants began the GSFMS. Each test was performed twice, with a 30-s rest between movements. Two examiners assessed the test results and recorded the raw scores. After completing the GSFMS test, the participants engaged in a standardised 10-min warm-up designed by the researcher. Standardised warm-ups include rod-free warm-ups (head movements, chest expansion exercises, abdominal and back exercises, body rotation exercises, knee joint movements, wrist and ankle exercises) and rod warm-ups (neck and shoulder mobility exercises, lateral body rotation with a bow step, and backswing rotational movements). All participants followed the same warm-up routine.

## GSFMS

The TPI Level 1 Screen (GSFMS) was used to assess functional movement patterns. It consisted of 16 items as listed in Table 2. The researchers demonstrated and implemented the GSFMS while providing standardised guidance. The movement tests were conducted in the same sequence as described by Rose (2003), with a 30-s rest interval between each movement. The participants performed each movement twice and the highest score was recorded. Participants who successfully completed a specific movement without pain during the tests received a score of 1; otherwise, they were scored 0. The two-sided test items were score as the sum of the left and right scores, and the maximum achievable composite score was 36 points.

## Golf performance

The testing criteria for golf performance were based on the standard of the students' sports skills grades (Ministry of Education of the People's Republic of China, 2019), which is a national standardized system for assessing the sports skill levels of primary, middle school and college students, covering 19 sports including soccer, track and field, golf, and others. Individual assessments (1-yard putt, 10-yard putt, 25-yard chip shot, 130/100-yard pitch, driving with a 1-wood) were first conducted.

(1) 1-yard putt: Putt five balls from a designated position 1 yard away from the hole, scoring two points for each ball successfully holed;

(2) 10-yard putt: Putt five balls from a designated position 10 yards away from the hole. Two points are scored for each ball that comes to rest within a circular area with the hole at the center and a radius of two yards;

(3) 25-yard chip shot: Use a wedge to chip five balls from a designated area 25 yards away from the hole. Points are awarded based on where the ball comes to rest within concentric circles around the hole: six points for a radius of five yards, seven points for

**Table 2 GFMS scoring criteria.**

| Item | GSFMS subitems | Scoring | | |
|------|----------------|---------|---|---|
| | | **2** | **1** | **0** |
| Unilateral test items (full score 10 points) | | | | |
| **PT** | Starting pelvic tilt | | Neutral tilt | S/C-Posture |
| | Amount of motion | | Normal motion | Both limited, hard time arching/flattening back |
| | Quality of movement | | Smooth movement | Shake and bake movement |
| **PR** | Without holding shoulders | | Good | Limited |
| | Coordination | | Good rotary movement | More lateral movement |
| **TR** | Without holding hips | | Good | Limited |
| **ODS** | Standing squat | | Bar overhead deep squat | Arms down full/limited deep squat |
| | Half kneeling ankle test | | Good dorsiflexion bilaterally | Right or left or both ankle dorsiflexion limited |
| | Do they weight shift? | | No weight shift | Weight shift right or left |
| **TT** | Bilateral toe touch | | Can | Can not |
| Two-sided test item (full score 26 points) | | | | |
| **90/90** | Standing | If greater than spine angle both sides | Right/left greater than spine angle | Equal to spine angle, less to spine angle |
| | Golf posture | If equal to standing | One-side equal to standing | Greater/less than standing |
| **SLB** | Thigh parallel | If 16–20 S both sides | Right/left 16–20 s | Other |
| **LT** | Low back flat against wall | If torso touches wall | One hand touches wall | No touches wall |
| **LQR** | Backswing | If 60 degrees or more | Right/left 60 degrees or more | Right and left less than 60 degrees |
| | Downswing | If 60 degrees or more | Right/left 60 degrees or more | Right and left less than 60 degrees |
| **STR** | Club behind back | If greater than 45 degrees | Right/left greater than 45 degrees | Other |
| **BWLE** | Lying supine | If glute normal both sides | Right/left glute normal | Other |
| **CR** | Mouth closed | If touches both sides | Touches right/left | Limited |
| **FR** | Elbows bent by sides | If both sides > 80 bilateral | Right/left > 80 bilateral | Palm UP/down limited |
| **WH** | Elbows bent by sides | If both sides normal | Right/left normal | Limited hinge up and down |
| **WF** | Bowing | If greater than 60 degrees both sides | Right/left greater than 60 degrees | Equal to 60 degrees, limited |
| **WE** | Cupping | If greater than 60 degrees both sides | Right/left greater than 60 degrees | Equal to 60 degrees, limited |
| Maximum composite score (36 points) | | | | |

**Note:**
PT, Pelvic Tilt Test; PR, Pelvic Rotation; TR, Test Torso Rotation Test; ODS, Overhead Deep Squat Test; TT, Toe Touch Test; 90/90, 90/90 Test; SLB, Single Leg Balance Test; LT, Lat Test; LQR, Lower Quarter Rotation Test; STR, Seated Trunk Rotation Test; BWLE, Bridge with Leg Extension Test; CR, Cervical Rotation Test; FR, Forearm Rotation Test; WH, Wrist Hinge Test; WF, Wrist Flexion Test; WE, The Wrist Extension Test.

4.5 yards, eight points for four yards, nine points for 3.5 yards, and 10 points for three yards;

(4) 130/100-yard pitch: With an iron, hit five balls from a designated area located 130 yards for men or 100 yards for women from the target. Points are awarded based on where the ball comes to rest within concentric circles around the target: six points

for a radius of 30 yards, seven points for 25 yards, eight points for 20 yards, nine points for 15 yards, and 10 points for 10 yards.

(5) Driving: Using a driver, hit five balls from a designated area towards the target located 150 or 120 yards within a 60-yard width for men or women, respectively. Points are awarded based on specific distance ranges:

① Scoring standard for men: 150–160 yards (five points), 160–175 yard (six points), 175–190 yards (seven points), 190–210 yards (eight points), 210–230 yards (nine points), and exceeding 230 yards (10 points);

② Scoring standard for women: 120–125 yards (five points), 125–130 yards (six points), for 130–150 yards (seven points), 150–170 yards (eight points), 170–190 yards (nine points), and for exceeding 190 yards (10 points).

(6) One round of 9-hole stroke play: This practical test involves playing nine holes on a golf simulator with a standard par of 36 and a maximum score of 30. The simulator system scores based on the actual number of strokes taken by the participant, providing the final score after completing nine holes.

Each assessment had a maximum score of 50 points. After completion of the individual assessments, groups of four participants engaged in practical assessments. The practical assessment has a maximum score of 30 points. The simulator displayed the hole-by-hole scores, total strokes, and overall scores. The mean total golf skill assessment score was 80 points.

## Statistical analysis

Data analysis was conducted using SPSS software (version 25.0; IBM, Armonk, NJ, USA). Various assessment indicators were expressed as means ± standard deviations (M ± SD). The normality of the data was assessed using the single-sample Kolmogorov-Smirnov test. For data that followed a normal distribution, independent-sample t-tests or one-way analysis of variance (ANOVA) tests were applied. In cases where the data did not conform to a normal distribution, the non-parametric Kruskal-Wallis test was used, followed by *post-hoc* testing with Bonferroni correction. The Spearman's correlation coefficient was used for non-normally distributed data with skewness, which is suitable for skewed datasets. The statistical significance level for the correlation coefficient (r) was set at $p < 0.05$, where $|r| < 0.4$ indicated weak correlation, $0.4 \leq |r| < 0.6$ indicated a moderate degree of correlation, and $|r| \geq 0.6$ indicated a strong correlation. Multiple linear regression analysis was used to determine how the individual GSFMS components (independent variables) predicted total golf skill score (dependent variable).

## RESULTS

### Correlation between GSFMS and golf performance

Figure 1 shows the relationship between the GSFMS scores and golf performance of all the participant data. Partial variables exhibited significant correlations, which were weak or
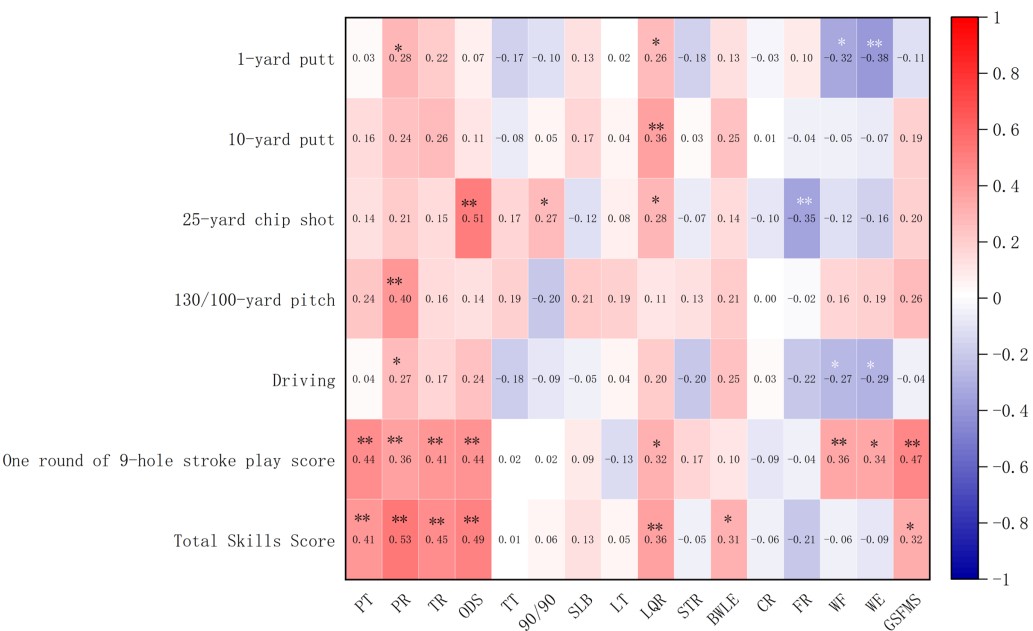

**Figure 1 Correlation analysis between GFMS and golf skills (r).** Note: *p < 0.05. **p < 0.01. PT, Pelvic Tilt Test; PR, Pelvic Rotation Test; TR, Torso Rotation Test; ODS, Overhead Deep Squat Test; TT, Toe Touch Test; 90/90, 90/90 Test; SLB, Single Leg Balance Test; LT, Lat Test; LOR, Lower Quarter Rotation Test; STR, Seated Trunk Rotation Test; BWLE, Bridge with Leg Extension Test; CR, Cervical Rotation Test; FR, Forearm Rotation (Pronation/Supination) Test; WF, Wrist Flexion Test; WE, Wrist Extension Test.

moderate. The results showed positive correlations between pelvic rotation and 1-yard putt ($r = 0.28$, 95% CI [−0.04 to 0.54], $p = 0.04$), 130/100-yard pitch ($r = 0.40$, 95% CI [0.18–0.59], $p = 0.001$), driving ($r = 0.27$, 95% CI [−0.03 to 0.52], $p = 0.04$), and one round of 9-hole stroke play score ($r = 0.36$, 95% CI [0.09–0.59], $p = 0.01$). Further, positive correlations existed between lower quarter rotation and 1-yard putt ($r = 0.26$, 95% CI [−0.08 to 0.57], $p = 0.01$) and 25-yard chip shot ($r = 0.28$, 95% CI [0.07–0.54], $p = 0.02$). However, wrist flexion was negatively correlated 1-yard putt ($r = −0.32$, 95% CI [−0.51 to −0.10], $p = 0.02$) and driving ($r = −0.27$, 95% CI= [−0.50 to 0.01], $p = 0.01$), as well as positively correlated with a round 9-hole stroke play score ($r = 0.36$, 95% CI [0.11–0.57], $p = 0.01$). Similarly, wrist extension was negatively correlated with 1-yard putt ($r = −0.38$, 95% CI [−0.56 to −0.18], $p = 0.001$) and driving ($r = −0.29$, 95% CI [−0.52 to −0.02], $p = 0.03$) and positively correlated with a single round 9-hole stroke play score ($r = 0.34$, 95% CI [0.10–0.56], $p = 0.01$). The Toe Touch Test (TT), the Single Leg Balance Test (SLB), the Lat Test (LT), the Seated Trunk Rotation Test (STR), the Cervical Rotation Test (CR), and the Wrist Hinge Test (WH) tests did not show statistically significant relationships with the golf performance indicators ($p > 0.05$).

## The impact of GSFMS on golf performance

Based on the significant correlation between the GSFMS and golf performance, to further clarify the extent of the impact of functional movement capabilities on golf performance, a multiple linear regression analysis was performed using the scores of each GSFMS

**Table 3 Multiple linear regression analysis of golf skill total score.**

| Predictor variable | $R^2$ | Adjusted $R^2$ | F | B | β | t |
|---|---|---|---|---|---|---|
| (Constant) | 0.50 | 0.31 | 2.66 | 35.58 | | 4.66* |
| PT | | | | 3.35 | 0.21 | 1.47 |
| PR | | | | 4.21 | 0.40 | 2.55* |
| TR | | | | 0.22 | 0.01 | 0.07 |
| ODS | | | | 1.08 | 0.12 | 0.81 |
| TT | | | | −2.36 | −0.10 | −0.74 |
| 90/90 | | | | −0.23 | −0.04 | −0.33 |
| SLB | | | | −0.72 | −0.04 | −0.30 |
| LT | | | | 1.35 | 0.08 | 0.65 |
| LQR | | | | 3.91 | 0.32 | 2.10* |
| STR | | | | −2.37 | −0.20 | −1.27 |
| BWLE | | | | 1.11 | 0.08 | 0.60 |
| CR | | | | 0.87 | 0.07 | 0.56 |
| FR | | | | −2.66 | −0.21 | −1.50 |
| WF | | | | −2.05 | −0.23 | −0.54 |
| WE | | | | 3.97 | 0.45 | 1.01 |

**Notes:**
* $p < 0.05$.
PT, Pelvic Tilt Test; PR: Pelvic Rotation; TR, Test Torso Rotation Test; ODS, Overhead Deep Squat Test; TT, Toe Touch Test; 90/90, 90/90 Test; SLB, Single Leg Balance Test; LT, Lat Test; LQR, Lower Quarter Rotation Test; STR, Seated Trunk Rotation Test; BWLE, Bridge with Leg Extension Test; CR, Cervical Rotation Test; FR, Forearm Rotation Test; WF, Wrist Flexion Test; WE, The Wrist Extension Test.

component item as an independent variable and the total golf skill score as the dependent variable. Table 3 shows that the pelvic and lower-body rotation abilities can significantly predict golf skill levels. These two variables collectively explained 31.2% of the variance in golf skill levels among collegiate golfers (Adjusted $R^2$ = 0.312; F = 2.663; $p < 0.05$). Standardised β values indicate that pelvic rotation has a more substantial impact on golf skill levels than lower-body rotation.

## DISCUSSION

This study using the GSFMS explored the relationship between functional golf movements and golf performance. The study found weak to moderate correlations between individual items in the skill test and GSFMS indicators. Further, we observed positive correlations between pelvic rotation and 1-yard putt, 130/100-yard pitch, driving, and one round of 9-hole stroke play score, which reflected the association between pelvic movement and the outcomes of these swing motions. Limited pelvic rotation ability may lead to excessive lateral movements during the swing, impacting the sequencing of the downswing and the separation of the upper and lower body (Kim et al., 2015). This could result in swing characteristics such as sliding, swaying, hanging back, as other parts of the body may compensate for the reduced pelvic rotation ability. Consequently, an optimal downswing posture may not be achieved, and unfavourable compensations in the kinetic chain could adversely affect the performance of motor skills. Furthermore, lower quarter rotation was positively correlated with 1-yard putt and 25-yard chip shot. These findings imply that the

rotation of the lower limb affects weight transfer and support, which is related to the effectiveness of the shots (*Gryc et al., 2015*). Thirdly, the negative correlation between wrist flexion and 1-yard putt and driving, as well as the positive correlation with a round 9-hole stroke play score, reflects the flexibility of the wrist. Similarly, wrist extension is negatively correlated with 1-yard putt and driving and positively correlated with a single round 9-hole stroke play score. These findings highlight the flexibility of the wrist. During wrist flexion and extension, corresponding skeletal movements occur (*Eschweiler et al., 2022*). If the wrist's flexion and extension abilities are restricted or excessive, it may lead to limitations in or excessive wrist cocking during the upswing, resulting in swing characteristics such as early casting scooping and over-the-top swing.

The GSFMS composite score was moderately positively correlated with the score of a single 9-hole round of stroke play ($r = 0.47$, 95% CI [0.23–0.65], $p < 0.01$), indicating a relationship between functional movement capabilities and golf performance. The higher the score in the tested 9-hole round of stroke play, the lower the handicap, which means that golfers with a lower handicap have higher levels of functional movement completion. This finding aligns with previous research. For example, *Keogh et al. (2009)* have found that golfers with lower handicaps performed better in flexibility tests of trunk rotation, wrist flexion/extension, and pelvic internal/external rotation. Additionally, *Speariett & Armstrong (2020)* have found a strong negative correlation between golfers' handicaps and the GSFMS composite scores.

We further found a weak positive correlation ($r = 0.32$, 95% CI [0.23–0.65], $p < 0.05$) between the total skill score and the GSFMS composite score, indicating a relationship between golf skills and functional movement capabilities. This corroborates findings from *Wu & Wang (2014)*, who have conducted a study using a self-selected indicator, the 150/120-yard pitch, as a skill test, and found a positive correlation between the flexibility and stability of the body's functional movement patterns and the level of specialised skills. *Huang et al. (2015)* have discovered that the total score for functional movements was significantly related to the level of specialised movements, which was primarily reflected in the performance of long-distance putts. *Lianpu (2015)* has found that training joint flexibility and stability can effectively improve ball striking results, and improving pelvic flexibility and strengthening the associated muscles can help increase clubhead speed, thereby increasing the hitting distance. These findings validated the assumptions of the Movement Pyramid model.

However, some studies found no relationship between functional movement and skill levels. *Parchmann & McBride (2011)* have conducted functional movement screening, 1RM (one-repetition maximum strength), and performance tests (10 and 20-m sprint times, vertical jump height, agility T-test time, and clubhead speed) on 25 NCAA Division I golfers. The study found that the total or individual scores on the functional movement screening did not significantly correlate with any of the performance test indicators. This result is consistent with the findings of *Okada, Huxel & Nesser (2011)*, in which total functional movement screening and individual scores did not have a direct relationship with any variables measuring performance. The findings of this study regarding the impact of functional movement capabilities on golf skill levels differ slightly from those of

previous studies. This difference may be due to the strong motor-skill level test method adopted in this study, whereas the previous test mainly focused on whole-body movement. In this study, two lower-body functional movement capabilities (pelvic and lower-body rotation) significantly predicted golf skill levels in golf major students, explaining 31.2% of the variance in golf skill levels. Pelvic rotation and lower-body rotation play a crucial role in the power chain of a golf swing (*Hume & Keogh, 2018*). The pelvis transfers energy from the lower limbs to the upper limbs and ultimately to the club, and lower-body rotation facilitates the generation of rotational torque, which is essential for generating clubhead speed and distance (*Choi et al., 2014*). Research has suggested that by focusing on the correct sequence and timing of pelvic and lower limb rotation, the power chain can be optimised, leading to improved athletic performance. However, incorrect or inefficient execution of these movements can place unnecessary strain on various parts of the body, resulting in injuries such as lower back pain (*Bourgain et al., 2022*). Therefore, the results of this study offer valuable insights for golf training and performance improvement. Coaches and golfers may prioritise training regimens that emphasise enhancing pelvic and lower-body rotation to improve golf skill levels. By integrating specific exercises targeting these functional movement capabilities, golfers can perform more efficient and powerful swings, thus enhancing their performance on the course. Addressing any deficiencies or limitations in pelvic and lower-body rotation will enable golfers to optimise their movement patterns, reduce injury risks, and ultimately achieve better overall performance and longevity in the sport.

The strength of this study lies in the use of a Performance Pyramid model to investigate the relationship between functional movements and sports performance. The assessment of sports performance predominantly focuses on objective measurements of athletic skills. However, this study is subject to certain limitations. While our primary objective was to investigate the predictive value of GSFMS on golf performance among collegiate golfers, it is essential to acknowledge that participant skill level and experience may have introduced variability in the results. Our study participants were collegiate golfers who were relatively new to the sport. Therefore, it is important to recognize that our findings may not be entirely generalizable to specific subgroups within the golfing population. Moreover, the evaluation of GSFMS involves a subjective assessment, which may introduce bias. Future research avenues could address these limitations by stratifying participants based on skill levels to provide a more nuanced understanding of the GSFMS's predictive capabilities in different proficiency groups. Additionally, the subjective assessment involved in evaluating the GSFMS warrants consideration. Future studies could benefit from adopting a more randomized sampling approach and incorporating motion analysis software to objectively identify swing characteristics. Furthermore, future research should explore the incorporation of various factors such as power and functional performance to predict and enhance athletic skill levels.

## CONCLUSIONS

This study found the weak to moderate correlations between the GSFMS and golf performance, and pelvic rotation and lower-body rotation abilities, thus predicting golf

skills. This GSFMS tool combined with skill assessment can help tailor training programs to address specific weaknesses and aid in injury prevention and improve overall performance.

In contrast, the present study employed the established GSFMS to comprehensively examine the relationship between functional movements and golf performance, contributing novel insights into the specific effects of functional abilities on overall skill and proficiency in the context of sports. This comprehensive approach sets this research apart from previous studies and enriches our understanding of the intricate relationship between functional movements and golf skills.

## ACKNOWLEDGEMENTS

The authors would like to thank the participants of this study. Additionally, the first author extends her appreciation to her coach Dr. Xianfei Wang and co-advisor Xiaoguang Li, for their dedicated efforts in training and mentoring her in the process of winning the honour of national level of golf player.

### Funding

This work was supported by the Hainan Province Philosophy and Social Science Planning Project (HNSK(YB)23-57). The funders had no role in study design, data collection and analysis, decision to publish, or preparation of the manuscript.

### Grant Disclosures

The following grant information was disclosed by the authors:
Hainan Province Philosophy and Social Science Planning Project: HNSK(YB)23-57.

### Competing Interests

The authors declare that they have no competing interests.

### Author Contributions

- Min Shi conceived and designed the experiments, performed the experiments, prepared figures and/or tables, and approved the final draft.
- Hua Wu conceived and designed the experiments, authored or reviewed drafts of the article, and approved the final draft.
- Hui Ruan performed the experiments, authored or reviewed drafts of the article, and approved the final draft.
- Dan Xu performed the experiments, prepared figures and/or tables, and approved the final draft.
- Libo Deng analyzed the data, prepared figures and/or tables, and approved the final draft.
- Shibo Pang analyzed the data, prepared figures and/or tables, and approved the final draft.

## Human Ethics

The following information was supplied relating to ethical approvals (*i.e.*, approving body and any reference numbers):

The Ethics Committee of the Hainan Provincial Sports Academy approval to carry out the study within its facilities (Ethical Application Ref No. GT-QM-02)

## Data Availability

The raw data are available in the Supplemental File.

## Supplemental Information

Supplemental information for this article can be found online at http://dx.doi.org/10.7717/peerj.17411#supplemental-information.

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
