# Peer review of "Using a golf specific functional movement screen to predict golf performance in collegiate golfers"

_PeerJ, doi:10.7717/peerj.17411_

## Round 0.1 · original submission · Major Revisions

The reviewers raised important points that need to be addressed in the revised manuscript. In particular, you should provide good explanations for the discrepancies between the raw data and the definition of scores (number of levels used for scoring) (see comments by Reviewer 3).

Reviewer 1 ·

Basic reporting

Generally well written, but a few areas where clear English could be improved. For example, lines 98 and 228 have grammar issues which appear to be from translation.

Figure 1: Clarify the difference between "total individual project score" and "total skills score" so this language is consistent with the manuscript. "Individual project score" does not appear within the manuscript body, so it is difficult to know what this refers to. Additionally, I appreciate the visual representation of these results, but the image appears pixelated. Ensure the resolution is appropriate. I also suggest changing the color scale so it encompasses all possible options for correlation values (1 to -1, rather than the current 0..6 to -0.4) to be more accurate.

Experimental design

Clear purpose statement and hypotheses. A few areas within the methods should be expanded or clarified:

1. Line 114-116 reports participants had never played golf prior to starting at the university. Do you have the number of years they have played golf, or how often? It appears these would be less skilled golfers with a relatively brief history of golf participation. If that is the case, this should be discussed within the context of your results. Skilled and experienced golfers may have very different results than those who just began to play the sport recently.

2. Line 145-148: For tests that would be completed bilaterally, clarify whether they scored separately for each side, averaged, or the best/worst score was used. Also, the authors report the movement must be completed without pain to receive a score of 1. Was it noted if individuals were able to complete the movement properly, but reported pain? This would likely lead to a different interpretation than someone who is unable to complete the movement, especially within your discussion (for example, where you state that excess wrist flexion/extension can affect putting performance: what if individuals receive a 0 because they reported pain but were able to achieve the desired range of motion?). If no individuals reported pain, I would recommend reporting this. Otherwise, this should be discussed.

3. Line 150-158: Provide much more thorough description of the golf performance testing, such as a brief description of what the assessments measured (such as distance, accuracy, etc. and the number of attempts for the putts, chip shot, etc.). It is hard to interpret your findings without a clear understanding of what was tested.

Validity of the findings

Some of the discussion points or conclusions need to be rephrased to appropriately match what was measured within the study and to not overstate study findings.

1. Line 203-207: Wrist flexion and extension components of the GSFMS simply test how much range of motion is available, not that an individual is hypermobile or unable to “maintain stability on the clubhead’s path”. Be careful to not make these assumptions. It is fair to hypothesize reasons why you found this negative correlation with the 1-yard putt but be clear that these are simply hypothesized explanations. Similar language exists in lines 218-221, and this should be restated as well.

2. Line 207-212: Soften the language used regarding technique, unless their technique was one of the measures that went into their final score. Otherwise, one can assume the authors did not objectively measure or analyze form, and it is not a guarantee that all golfers followed the same technique (left and club shaft in a straight line, no forearm rotation). Rephrase this statement to be clear that this is the desired technique, which is why you likely found this correlation, not that all golfers followed this technique exactly.

·

Basic reporting

There are no issues with basic reporting. English writing is good and to a standard where no major improvements are required.

Experimental design

The general design is robust and the methods are detailed enough to allow for replication. The design is also sufficient to answer the question at hand. However, there are a few places for clarification (please see additional comments).

Validity of the findings

While the interpretation of findings can be improved with the addition of confidence intervals and a few other points (see additional comments), the validity of the findings appear strong.

Additional comments

Title:
The title is fine, but I think adding “the validity of” or “Using a”, or something similar to the very beginning would improve clarity. I.e., “Using a golf specific functional movement screen to predict golf performance in collegiate golfers”

Abstract:
Good abstract.
The number of significant digits in the standard deviations can be reduced by one.
Can the authors better explain ‘partial’ variables? Are these simple correlations (i.e., two variables correlated with one another)?
If the word limit allows, it would be great to know what the standardized beta-values for the pelvic rotation and lower-body rotation in the last sentence of the results.

Introduction:
Nicely done! The authors set the study up well and provide good justification for the importance of their study and the gaps they aim to address.
My main criticism is that the final paragraph is unnecessarily long and should be edited for brevity.

Methods:
How/what software was used to perform the sample size estimation?
Might there be a source for the warm-up movements? Is this from Gray Cook or Titlist?
The statistical tests are logical and follow the correct order. My main suggestion would be to include 95% confidence intervals for the correlations to make it easier for the reader to interpret the spread of the data/results. These 95%CIs would only need to be reported in-text for the significant correlations.
The authors should also be clear how the correlational results are to be interpreted (e.g., <0.10=trivial, 0.10-0.30=small etc.)

Results:
As mentioned for the methods, I suggest reporting 95%CIs for the statistically significant correlations.

Discussion:
While well written, the discussion is lacking in a few important ways.
Firstly, the first paragraph should BRIEFLY re-introduce the aims and methods, then BRIEFLY summarize the most important findings. The information following the first 2 sentences should be in its one separate paragraph. Additionally, some actually numerical data would help make the introduction clearer. Instead of simply stating ‘…which is why writs flexion and extension are negatively correlated with the 1-yard putt.’ It would be great to include the r- and p-values.

Tables/figures:
Like the abstract, please reduce the number of significant digits for the standard deviation in table 1.
Table 2 can be re-formatted to fit on a single page for easier reading. This might be more of a post acceptance issue, but it would make it easier to review etc. Also, the table should be understandable on its own. So, either spell out the abbreviations in full, or define the abbreviations in the table legend at the bottom.
Big fan of the heat-map figure. Great way of illustrating many findings at the same time.

·

Basic reporting

This study sets out to examine the relationship between athletes’ abilities to perform functional movement and golf performance. Overall, the structure of the paper is clear and the language concise.

The Introduction is comprehensive, however the Methods section is missing some details (see below), the Result section is brief and the Discussion section would benefit from a more in-depth consideration of the results and possible interpretations.

Thank you for providing the raw data. However, both for golf performance and functional screening it is unclear to me how scores were applied: For golf performance, scores for individual tasks appear to be on a scale from 0 to 10, but it is unclear how they were calculated (e.g. distance to target? Subjective?). For the functional tests, scoring criteria are only provided for levels “1” and “0” for many of the tasks, but several participants achieved the undefined level “2” (for example for the Pelvic Tilt Test). In addition to this, athletes achieved higher scores than what is defined in Table 2 (e.g. level “3” in Pelvis Tilt (participant 55) or even level “4” in the 90/90 test (several participants). I suggest adding definitions of the golf scores and a clarification of the functional movement scores.

Experimental design

You summarise four previous studies presenting relationships between golf performance and functional movement screening. While none of them uses the same approach as the present paper, one could argue that some of the previous papers use more specific and established measures of golf movement quality such as swing sequence and measure golf performance more directly through (for example) ball speed and accuracy. This section would benefit from more detailed arguments as to how the current study fills a knowledge gap.

Also, please provide information about the definition of the golf scores so that statements about the rigorousness of the experimental approach can be made.

Validity of the findings

I have two main concerns regarding the validity if the findings: (1) Lack of consideration of the complexity of motor control when performing a golf swing; (2) Implication of causal relationships when interpreting correlations and


Regarding (1), you claim that increased flexibility of the wrist has a negative effect on putting performance (line 203-205). However, putting is a low intensity task and I find it reasonable to expect that a person with high flexibility of the wrist could still maintain a stable wrist through muscle forces and motor control. The same applies to your statement regarding chipping performance and forearm rotation (line 211); again, a skilled golfer would be able to maintain a stable position in this low intensity task, no matter how flexible they are.

Regarding (2), for example you state in the abstract (line 33) that you provide insights in the *effects* of functional abilities. However you can only make statements about the relationship of parameters, Similarly, in line 216 you state that the sequence of body rotation *explains* the significant positive correlation between pitching performance and pelvic rotation. I suggest carefully rewording the Abstract, Discussion and Conclusion to make sure that this study simply observes correlations and does not establish causal relationships.

In addition, please include all statistically significant correlations in the discussion or make a rationale for only including some.

Additional comments

Line 41: You mention “in the past 5 years”, but the data appears to be from 2021. Suggest mentioning the time span for when the data was valid.
Line 116: (1) Inability to participate in golf – for which reasons?
Line 181 Cannot find Wrist Hinge Test (WH) in Figure 1. Is it missing or is different terminology used?
Line 228 Consider replacing “fewer handicaps” with “lower handicap”

---

## Round 0.2 · Major Revisions

The reviewers acknowledged improvements in the manuscript. However, there remain some considerable issues that need to be addressed.

Reviewer 1 ·

Basic reporting

Thank you for clarifying the participants' golf experience as well as improving the description of the methods used. The manuscript could still benefit from proofreading for grammar and language issues to improve the quality of the writing (some examples: lines 87-95, 106-107, 128-130, 162-164, 220-233, 251-254, 260-261, 266-267, 269-270).

Experimental design

Thank you for providing more detail regarding the methodology. This is much improved and allows for replication.

Statistical analysis section does not seem to match with the reported results. For example, lines 204-209 discuss comparing means, but it is unclear where this was done. The results only report correlations and a regression analysis. The only mention of means (or any raw data) is within table 1 of a few demographics. I also don't think these need to be split by sex as this was not a purpose of the study.

Issues with the regression analysis aspect of the research question are below.

Validity of the findings

Now that there is sufficient detail within the methods to understand what tests comprised the golf performance outcomes, it is important to report the participants' performance on these tests. A table or figure should be included to report means, standard deviation, etc. of each of the components of the golf performance outcomes (such as avg 1-yard putt score, avg 10-yard putt score, etc.). Currently only relationships between variables are reported, which severely limits the impact of this manuscript.

The regression analysis needs further justification and/or discussion in order to add value to the manuscript. With 16 independent variables included to predict performance on the golf skills assessment for 56 participants, this may be under-powered, especially with the finding that only 2 variables were significant predictors and explained just 31% of the variance in performance. Either the regression analysis should be better justified within the purpose statement and discussion, or I recommend it should be removed. In line 213-215 you state the purpose of the regression was to determine the relationship between GSFMS components and skill score, but that is already established with your correlation analysis. Regression should be used to predict performance - therefore you should discuss the potential value for coaches, golfers, clinicians, etc. in using the pelvic rotation & lower quarter rotation as significant predictors of golf performance.

Additional comments

Lines 220-236 seem repetitive with Figure 1. I understand another reviewer has requested further statistical information such as 95% CI, which is likely why these lines were included, but it seems that this could be represented more effectively.
I appreciate the figure and believe it is a helpful visual, so I would recommend moving the statement from lines 220-222 to the end of this section, with the text more clear that the figure is meant to be a visual supplement to the more complete results reported in-text.

·

Basic reporting

Excellent, no issues

Experimental design

No issues

Validity of the findings

No issues

Additional comments

The authors have done a good job of addressing my comments/concens.

I have no other issues to be resolved.

·

Basic reporting

Raw data - golf and functional movement scores: Thank you for adding definitions of the scores and for explaining that scores from left and right side were aggregated. This explains the numbers in the raw data, so I am satisfied with this point.
No further comments.

Experimental design

Thank you for adding definitions of the golf scoring system. This raises some follow-up questions:
1. Were all golf tests performed with the simulator, even the short putting tests (1) and (2)?
2. Is it correct that scores for the driving test (5) were only based on distance, not accuracy/direction?
3. Some of the scoring levels require a resolution of the predictive capabilities of 5 yards and even 0.5 yards (Chipping). Can you provide any information about the accuracy of the Q9 Golf Simulator? If no accuracy studies have been performed, could you add a short description of the applied tracking technology (optical, radar, ...)?

Validity of the findings

Thank you for addressing my feedback - no further comments.

Additional comments

(Line numbers refer to "Tracked changes" Word document)

LN 123-127: I suggest restructuring this sentence - it was unclear to me to which part the "using the national standard..." part relates to. It could help to break it down into two sentences: "This study aimed to deepen our understanding of the relationship between functional movement and overall golf performance." "GSMFS was used to assess functional movements, and the national standard of student sports skill rating by age and sport (Ministry of Education of the People’s Republic of China)"

---

## Round 0.3 · Minor Revisions

You have given due consideration to all reviewer comments. Before I can make a final decision, I request some minor changes. Please see the attached PDF file for more details

---

## Round 0.4 · accepted · Accept

All comments have been adequately addressed and the paper is ready for publication. I would like to congratulate the authors on their work.